# Modelling the impact of changes to abdominal aortic aneurysm screening and treatment services in England during the COVID-19 pandemic

**Lois G. Kim**[1]*, **Michael J. Sweeting**[1,2], **Morag Armer**[3], **Jo Jacomelli**[3], **Akhtar Nasim**[4], **Seamus C. Harrison**[1]

1 Cardiovascular Epidemiology Unit, Department of Public Health and Primary Care, Strangeways Research Laboratory, University of Cambridge, Cambridge, United Kingdom, 2 Department of Health Sciences, University of Leicester, George Davies Centre, Leicester, United Kingdom, 3 Public Health England, Wellington House, London, United Kingdom, 4 Sheffield Teaching Hospitals NHS Foundation Trust, Northern General Hospital, Sheffield, United Kingdom

* lois.kim@medschl.cam.ac.uk

**Data Availability Statement:** The DES model used in this work is available on a GitHub repository

## Abstract

### Background

The National Health Service (NHS) abdominal aortic aneurysm (AAA) screening programme (NAAASP) in England screens 65-year-old men. The programme monitors those with an aneurysm, and early intervention for large aneurysms reduces ruptures and AAA-related mortality. AAA screening services have been disrupted following COVID-19 but it is not known how this may impact AAA-related mortality, or where efforts should be focussed as services resume.

### Methods

We repurposed a previously validated discrete event simulation model to investigate the impact of COVID-19-related service disruption on key outcomes. This model was used to explore the impact of delayed invitation and reduced attendance in men invited to screening. Additionally, we investigated the impact of temporarily suspending scans, increasing the threshold for elective surgery to 7cm and increasing drop-out in the AAA cohort under surveillance, using data from NAAASP to inform the population.

### Findings

Delaying invitation to primary screening up to two years had little impact on key outcomes whereas a 10% reduction in attendance could lead to a 2% lifetime increase in AAA-related deaths. In surveillance patients, a 1-year suspension of surveillance or increase in the elective threshold resulted in a 0.4% increase in excess AAA-related deaths (8% in those 5–5.4cm at the start). Longer suspensions or a doubling of drop-out from surveillance would have a pronounced impact on outcomes.

(https://github.com/mikesweeting/AAA_DES_
model).

**Funding:** This work was supported by core funding
from: the UK Medical Research Council (MR/
L003120/1), the British Heart Foundation (RG/13/
13/30194) and the NIHR Cambridge Biomedical
Research Centre (BRC) [The views expressed are
those of the author(s) and not necessarily those of
the NIHR or the Department of Health and Social
Care]. LGK is funded by the NIHR Blood and
Transplant Research Unit in Donor Health and
Genomics (NIHR BTRU-2014-10024). SCH is
funded by an MRC CARP Fellowship (Mr/T023783/
1). https://mrc.ukri.org/ https://www.bhf.org.uk/
for-professionals https://cambridgebrc.nihr.ac.uk/
http://www.donorhealth-btru.nihr.ac.uk/. The
sponsors and funders played no role in study
design. data collection, decision to publish or
preparation of the manuscript.

**Competing interests:** No authors have competing
interests.

## Interpretation

Efforts should be directed towards encouraging men to attend AAA screening service
appointments post-COVID-19. Those with AAAs on surveillance should be prioritised as the
screening programme resumes, as changes to these services beyond one year are likely to
have a larger impact on surgical burden and AAA-related mortality.

## Introduction

In March 2020 the initiation of the nationwide "lockdown" to protect against transmission
of COVID-19 had a profound effect upon the delivery of routine services provided by the
UK National Health Service (NHS). This included a substantial reduction in the number of
cardiovascular procedures performed including repair of Abdominal Aortic Aneurysms
(AAA) [1]. Furthermore, AAA screening (including surveillance) in most areas of the UK
was paused during the lockdown due to concerns about COVID-19 transmission and
whilst strategies were considered regarding mitigating risk from delayed surgical interven-
tion (as a result of reduced capacity within the NHS). Ruptured AAA carries a high mortal-
ity [2] and screening for AAA is offered to men in their 65th year throughout England via
the NHS Abdominal Aortic Aneurysm Screening Program (NAAASP) [3]. Those with
small and medium AAA (3.0–5.4 cm) are offered ultrasound surveillance quarterly or
annually depending on size, whilst those with large AAA ($\geq$5.5 cm) are referred for consid-
eration of elective surgical repair, before the risk of rupture becomes too high. Circa
300,000 men are offered screening annually, of whom around 1% are found to have an
AAA [4], whilst approximately 15,000 men are currently under surveillance in the
programme.

During the first lockdown the UK National Joint Vascular Implementation Board [5]
suggested that in individuals with AAA measuring 5.5–6.0 cm elective surgery could be
delayed for up to 12 months, and those 6.0–7.0 cm for up to 6 months. The capacity to offer
elective AAA repair has been severely reduced and the number of elective AAA repairs dur-
ing the lockdown period fell dramatically to around 12% of pre-COVID procedures per
week in April 2020 [6], with a larger proportion >7 cm than pre-COVID-19 (20% versus
10% for elective infra-renal repairs) [6]. Furthermore, nosocomial transmission of
COVID-19 (and associated excess mortality) has been reported throughout the UK [7]
even after introduction of COVID-minimised pathways (so called GREEN pathways) to
reduce the risk of this happening. This has led to a backlog of men awaiting elective sur-
gery, a backlog of unscreened 65-year-old men, and men with known AAA in surveillance
who may have gone over the referral threshold. As further waves of COVID-19 infection
grip the UK there are uncertainties as to how to manage this situation with a delicate bal-
ance between risk of COVID-19 transmission during screening and treatment and compet-
ing demands on NHS resources balanced against the risk of untreated AAA rupturing. This
is particularly stark when the large number of new cases thought to be due to a more trans-
missible variant is considered, especially as those with AAA are likely to be at high risk of a
poor outcome following COVID-19 infection as there appear to be shared risk factors for
both [8].

Here, we use a discrete event simulation model for AAA screening, previously developed
and validated to provide evidence for the national screening programme for men in England,
to explore different approaches to post-lockdown service resumption.

## Methods

### Model

NAAASP invites men at age 65 for an ultrasound scan [3]. Those with an aortic diameter <3cm are considered at acceptably low future risk of growth and rupture and are discharged. Those with small AAAs (3.0–4.4 cm diameter) are recalled for ultrasound-based surveillance on an annual basis, and those with medium AAAs (4.5–5.4 cm diameter) every three months. Whilst under surveillance there is a risk that individuals may suffer a ruptured AAA, with the risk of rupture increasing as a function of the size of the aneurysm [9]. Once an AAA reaches 5.5 cm in size, the risk is deemed too great and the individual is referred for a consultation to consider elective repair. These features define the screening programme policy, though at any stage an individual may choose to decline screening or surveillance. It is the policy of the programme to not further follow-up those who decline at any stage, though such individuals may subsequently have incidental detection (or re-detection) of their AAA following investigations outside of the screening programme.

A discrete event simulation (DES) model has previously been developed in R version 3.6.3 [10] to represent the life course of individuals invited to AAA screening and to allow investigation of different screening policies without the need for large clinical trials [11]. The full pathway reflecting both the natural history and screening programme is described in detail elsewhere [10, 11]. In brief, transitions between each state are modelled using rates and probabilities informed by NAAASP, and large randomised trials and studies of AAA screening and surgery (S1 Table). These parameters are used to simulate future event times for key screening and clinical events such as AAA rupture alongside different screening and intervention policies. Here, we recommission this model to investigate the potential impact of changes to NAAASP in the light of COVID-19.

The original DES model simulated events for a new cohort at a given age (e.g. 65) from the time of invitation to screening up to their date of death or age 95 (the time horizon). The repurposed DES model is extended here to allow events to be simulated from a cohort of individuals already under surveillance in the NAAASP, through simulation of key characteristics (age and aortic diameter) at the inception of the model ("time zero"), which is taken to be March 2020 when the initial UK national "lockdown" was imposed. The model is further extended to allow temporary suspension of a) invitation to primary screening, b) surveillance appointments, c) elective surgery, and to allow transitory increases in a) the AAA diameter threshold for elective surgery and b) non-attendance at both primary screening and surveillance scans (which may arise due to individuals engaging in more cautious behaviours as a result of the pandemic [12]). The full model and data underpinning the population initialisation can be found on a GitHub repository at https://github.com/mikesweeting/AAA_DES_model.

### Data

We are interested in outcomes for two different cohorts affected by changes to NAAASP services: (1) men turning age 65 who are eligible for the primary screening scan from March 2020 onwards, and (2) men with known AAAs under surveillance within the national screening programme in March 2020. For the surveillance cohort, the characteristics of the modelled population (joint distribution of age and AAA diameter) are informed from data supplied by NAAASP about the population of men under surveillance in May 2020 (personal communication).

## Post-COVID-19 policy scenarios

Public Health England and the UK National Joint Vascular Implementation Board published guidelines for the resumption of AAA screening and intervention services respectively in June 2020 [5, 13]. In the first instance, we explore the impact of the proposed post-COVID-19 changes to services in a series of analyses altering each aspect of the screening and intervention process in turn, over a range of potential transition period lengths. The model is broken up into three periods, with the second and third periods differing in length over the different scenarios modelled:

1. Full lockdown period (months 1–3; corresponds to Apr-Jun 2020 in the UK)

2. Transition period (subsequent period of reduced service, ranging from 0-60m in length depending on scenario; corresponds to Jul 2020 onwards in the UK)

3. Return to pre-COVID-19 service (post-transition period)

The full set of scenarios modelled is detailed in Table 1. We investigate the impact of changes relating to the invited group (varying suspension of invitation and attendance rates) and to the surveillance cohort (varying suspension of surveillance scans, drop-out rates, and time at increased size threshold for elective surgery) in a univariate manner. In addition, the cumulative impact of these changes in the surveillance cohort is also assessed.

Each scenario model is run for 10 million hypothetical individuals randomly drawn with replacement from the distribution of the relevant population. The models are run for a period of 30 years, with screening policies reverting to pre-COVID-19 norms for the whole of the

**Table 1. Modelled COVID policy scenarios for AAA services.**

| Cohort | Model | Status quo model parameters* |
|---|---|---|
| Invited 65-year-olds | I0 (status quo) | Attendance: 75% |
| | | Drop-out rate/annum: 6% |
| | | Threshold for surgery: 5.5 cm |
| Surveillance | S0 (status quo) | Drop-out rate/annum: 6% |
| | | Threshold for surgery: 5.5 cm |
| | | **Changes from status quo model**** |
| Invited 65-year-olds | I1 | Delay to invitation: ranges from 3 months– 5 years |
| | I2 | Delay to invitation: 6 months |
| | | Attendance: 45–75% |
| Surveillance | S1 | Suspension of surveillance scans: ranges from 3 months– 2 years |
| | S2.1 | Suspension of elective surgery: 3 months |
| | | Suspension of surveillance scans: 6 months |
| | | Drop-out rate/annum: ranges from 6–15% for 1 year |
| | S2.2 | Suspension of elective surgery: 3 months |
| | | Suspension of surveillance scans: 6 months |
| | | Drop-out rate/annum: ranges from 6–15% for 2 years |
| | S3 | Suspension of elective surgery: 3 months |
| | | Suspension of surveillance scans: 6 months |
| | | New threshold for surgery: 7.0 cm for 6 months up to 5 years |

* Full details regarding parameter inputs and sources provided in S1 Table.

** Unlisted parameters remain unchanged from status quo. All time periods listed relate to the period following March 2020.

post-transition period. Model convergence is summarised using cumulative results from consecutive sub-runs each of 1 million individuals (S1 and S2 Figs). Total numbers of AAA-related deaths, operations (both elective and emergency) and ruptures over the whole follow-up period are recorded for each model. These clinical results are reported as percentage change from the status quo as well as expected increase in number of events when scaled to the population of England [14].

## Results

### Invited 65-year-old cohort

Fig 1 shows the impact on outcomes following different delays to invitation to primary screening amongst current 65-year-olds. Changes to outcomes are small in the first two years of delay, though it is estimated there could be a 4% increase in AAA-related deaths over the lifetime of this cohort if delays in invitation continued for five-years, resulting in 120 additional AAA-related deaths (Table 2). Changes in attendance rates have a more marked impact on outcomes, where reduced attendance could result in 124 (4.5%) more AAA-related deaths and 66 (6.0%) additional emergency operations if attendance dropped from 75% to 55% (Fig 2, Table 2).

### Surveillance cohort: Scan suspension

Suspending ultrasound scans in the surveillance cohort could result in 9 (0.4% increase) additional AAA-related deaths if scans were suspended for one year (Table 3, Fig 3). Of these, 2

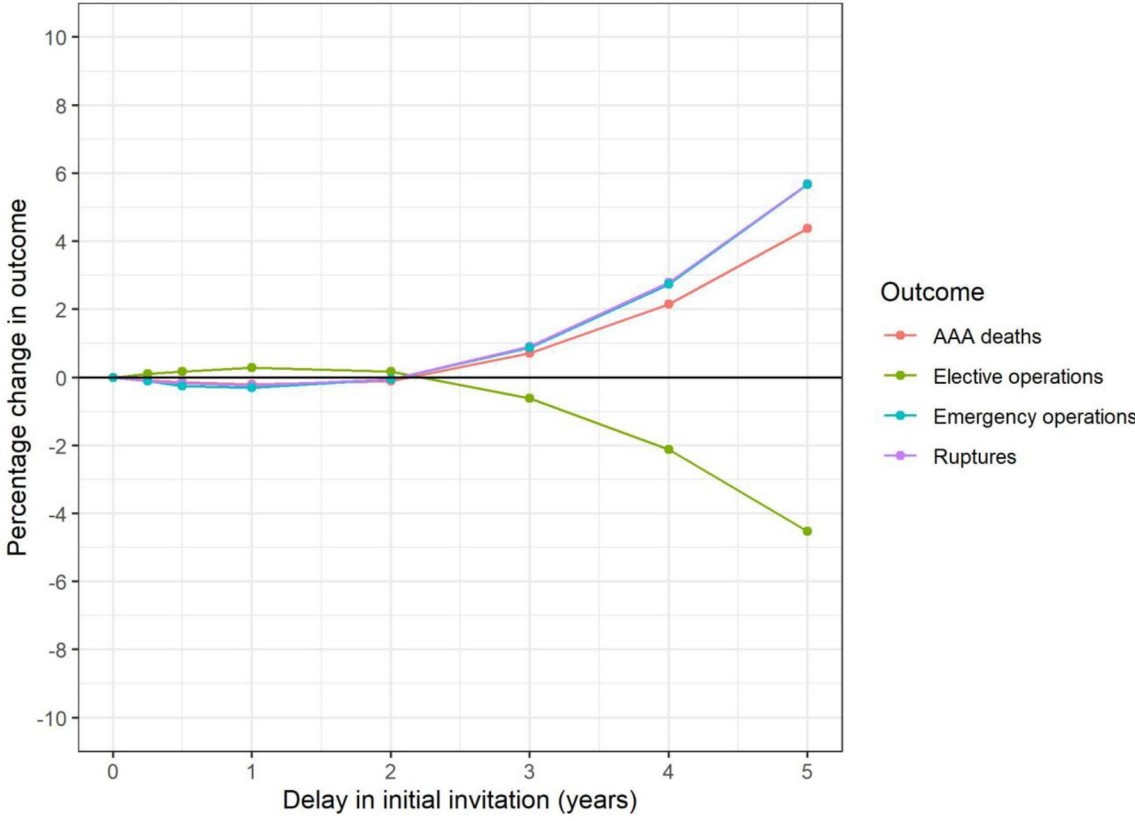

**Fig 1. 65-year-old cohort: Change in key outcomes over varying delay to primary invitation (model I1).**

**Table 2. Predicted excess AAA deaths and emergency operations in the national invited 65-year-old cohort over 30y period.**

| Length of delay to invitation | Excess AAA deaths (excess emergency operations) in Model I1* | Attendance rate at primary scan | Excess AAA deaths (excess emergency operations) in Model I2* |
|---|---|---|---|
| 6m | 0 (0) | 65% | 56 (29) |
| 12m | 0 (0) | 55% | 117 (62) |
| 24m | 0 (0) | 45% | 175 (94) |
| 36m | 18 (9) | | |
| 48m | 55 (29) | | |
| 60m | 112 (59) | | |

* Model I1 = delay to invitation; Model I2 = reduced attendance at primary scan (with 6m delay to invitation)

Notes: National male 65-year-old cohort for England: n = 279,798; expected AAA deaths over 30y in status quo = 2564; expected emergency operations over 30y in status quo = 1041.

(1% increase) are in the sub-group measuring 4.5–4.9 cm at the start of the pandemic and 7 (8% increase) in the sub-group measuring 5.0–5.4 cm; <0.1 are in the 3.0–4.4 cm sub-group. More pronounced effects are evident for suspension for two years and beyond. Suspending surveillance for two years could result in 40 excess AAA-related deaths overall; a 1.9% increase over the lifetime of the surveillance cohort. Of these, 1 is in the 3.0–4.4 cm sub-group and 17 (7% increase) in the 4.5–4.9cm sub-group. However, the remaining 22 excess deaths are in the 5.0–5.4cm range, corresponding to a 24% increase in AAA-related deaths in this sub-group.

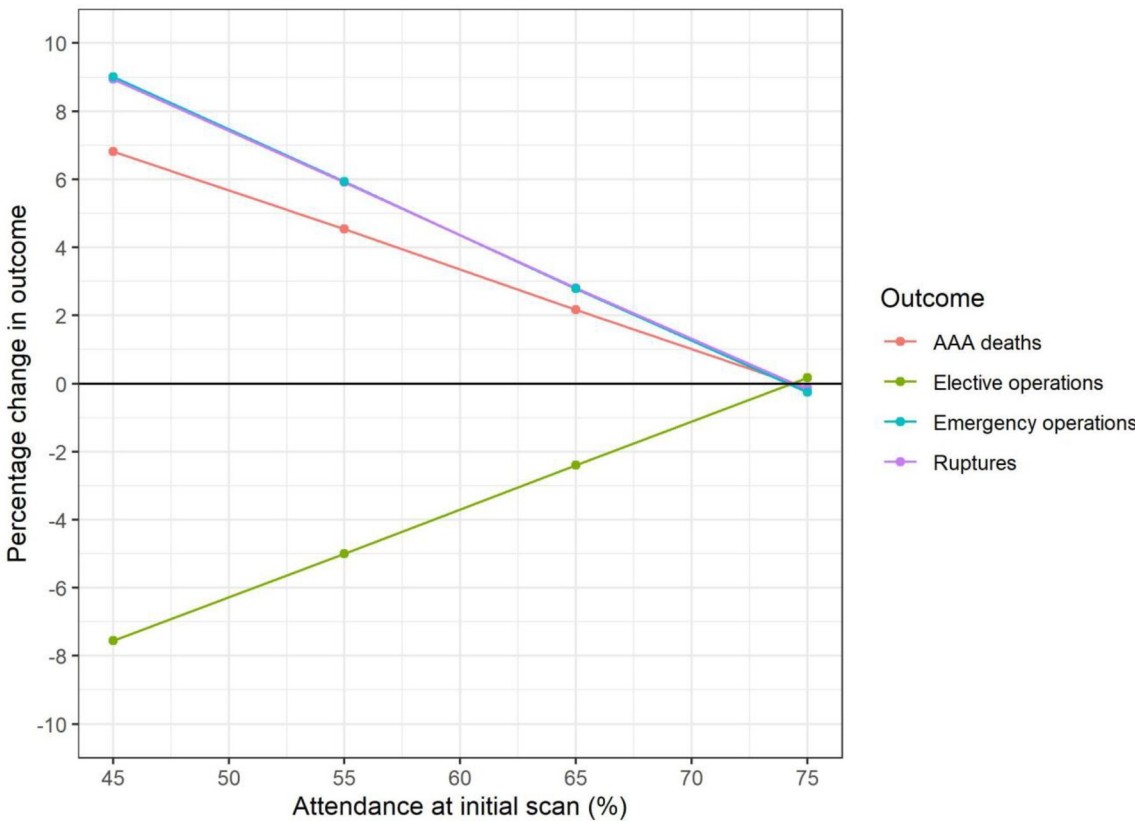

**Fig 2. 65-year-old cohort: Change in key outcomes over varying attendance at primary scan (model I2).**

**Table 3. Predicted excess AAA deaths and emergency operations in the national surveillance cohort over 30y period.**

| Length of scan suspension | Excess AAA deaths (excess emergency operations) in Model S1* | Length of time at 7cm threshold | Excess AAA deaths (excess emergency operations) in Model S3* | Dropout rate/ annum | Excess AAA deaths (excess emergency operations) | |
|---|---|---|---|---|---|---|
| | | | | | Model S2.1* | Model S2.2* |
| 6m | 2 (1) | 6m | 2 (1) | 8% | 46 (24) | 84 (44) |
| 12m | 9 (5) | 12m | 9 (5) | 10% | 84 (44) | 152 (81) |
| 24m | 40 (22) | 24m | 41 (23) | 12% | 120 (63) | 219 (116) |
| 36m | 110 (61) | 36m | 99 (55) | 15% | 174 (92) | 314 (167) |
| 48m | 230 (126) | 48m | 176 (98) | | | |
| 60m | 408 (222) | 60m | 260 (146) | | | |

* Model S1 = scan suspension; Model S3 = 7cm threshold for elective surgery Model; Model S2.1 = increased dropout for 1y; Model S2.2 = increased dropout for 2y

Notes: National surveillance cohort in March 2020: n = 15,376; expected AAA deaths over 30y in NAAASP status quo = 2152; expected emergency operations over 30y in NAAASP status quo = 745.

## Surveillance cohort: Drop-out

In scenarios where the drop-out rate (incorporating declined appointment, out of cohort and non-attendance due to medical reasons) is increased, there is a considerable impact on clinical outcomes (Fig 4). Currently in NAAASP the drop-out rate among those under surveillance is around 6%/annum, though recent research has suggested as few as 59% of invitees may be

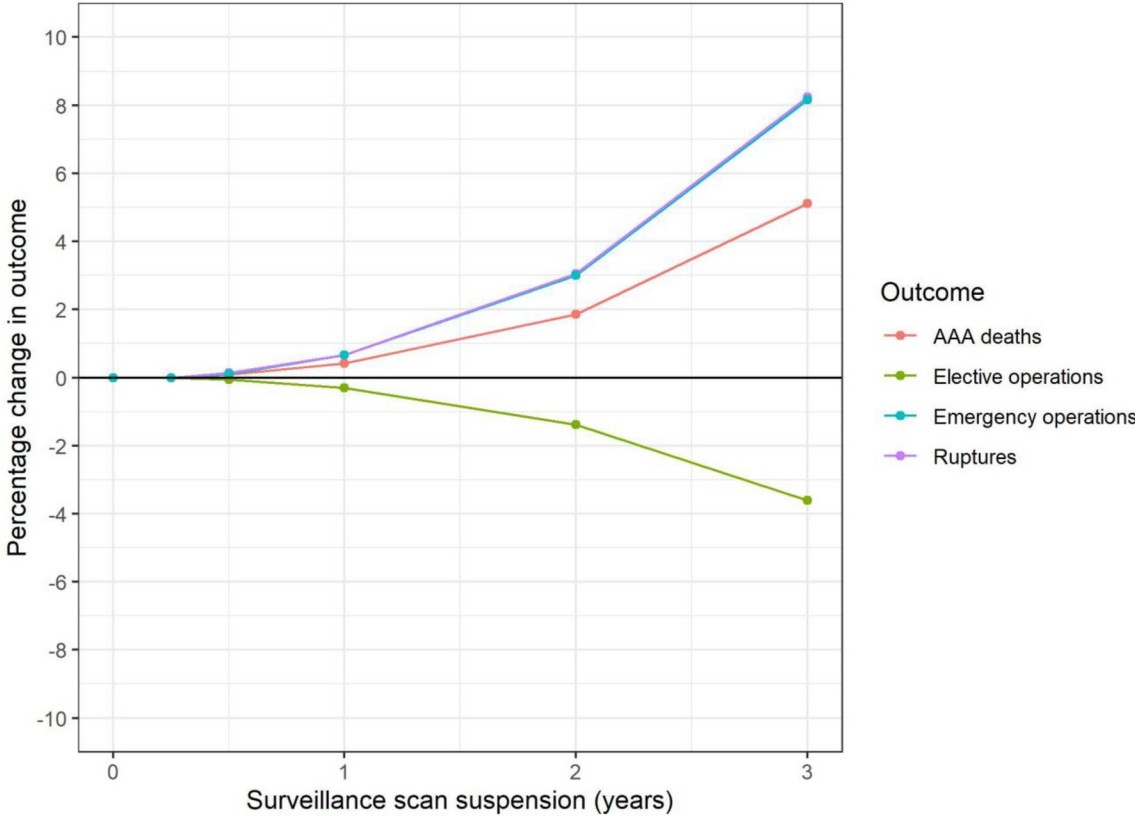

**Fig 3. Surveillance cohort: Change in key outcomes over varying suspension of surveillance scans (model S1).**

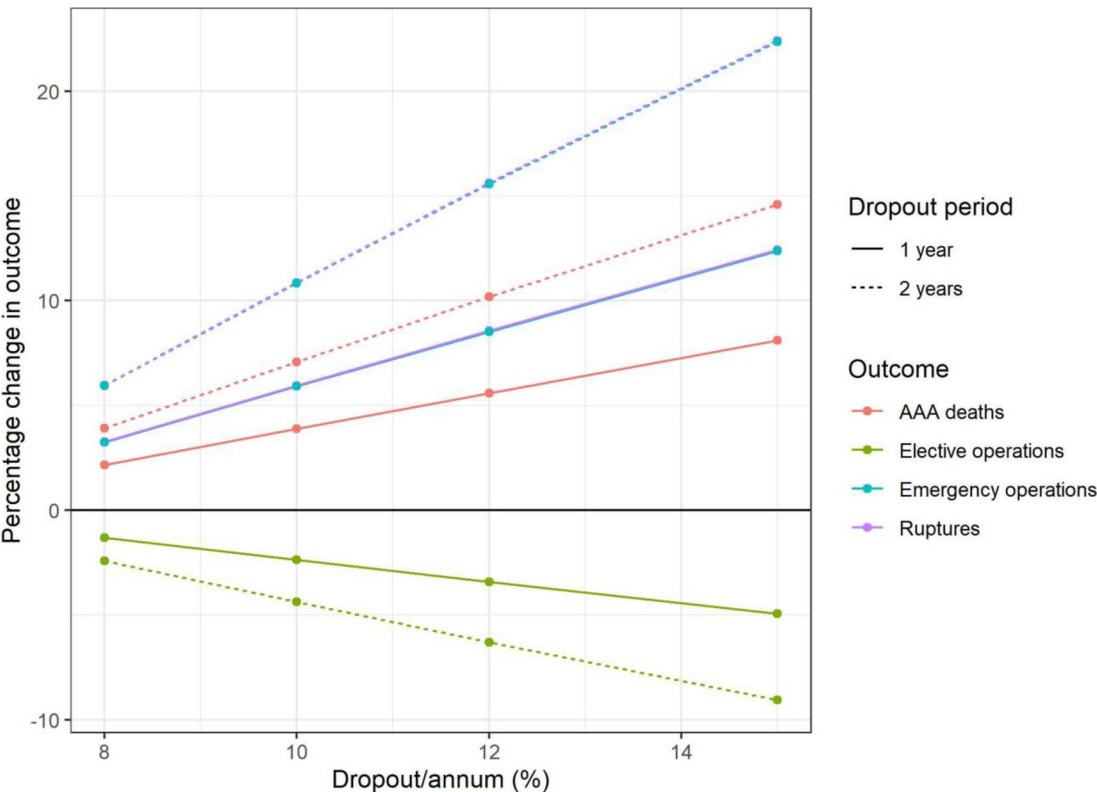

**Fig 4. Surveillance cohort: Change in key outcomes over varying dropout rates, applied for (i) 1y (model I2.1) and (ii) 2y (model I2.2).**

willing to attend a follow-up appointment during the pandemic [12]. AAA-related deaths could be expected to increase by 84 deaths (3.9%) over the lifetime of the cohort if this drop-out rate increased to 10%/annum for the first year of the pandemic, or 152 (7.1%) excess AAA-related deaths if this increased drop-out continued for two years (Table 3). At 15%/annum drop-out over the course of a year there was an estimated 8.1% increase in AAA-related deaths, resulting in 174 additional AAA-related deaths, and a 14.6% increase if it continued for two years. There is a greater estimated impact on ruptures and emergency operations, which are estimated to increase by around 6% with a 10%/annum drop-out rate for one year (11% for two years) and by 12% with a 15%/annum drop-out rate for one year (22% for two years).

## Surveillance cohort: Threshold for surgery

Fig 5 and Table 3 show the impact of increasing the size threshold for elective surgery to 7cm and varying how long this threshold is applied. There is a modest increase of around 9 (0.4%) AAA-related deaths over the lifetime of this cohort if the increased threshold is applied for one year. However, if the threshold is increased for two years, AAA-related deaths are estimated to increase by 41 (1.9%). Ruptures (and consequently emergency operations) follow a similar pattern, with around a 0.7% increase for one year at the increased threshold, rising to 3.1% for two years.

## Surveillance cohort: Cumulative impact of changes

S3 Fig shows the results of sequentially adding service changes in order to assess the cumulative impact of these scenarios on AAA-related deaths in the surveillance cohort. S2 Table and

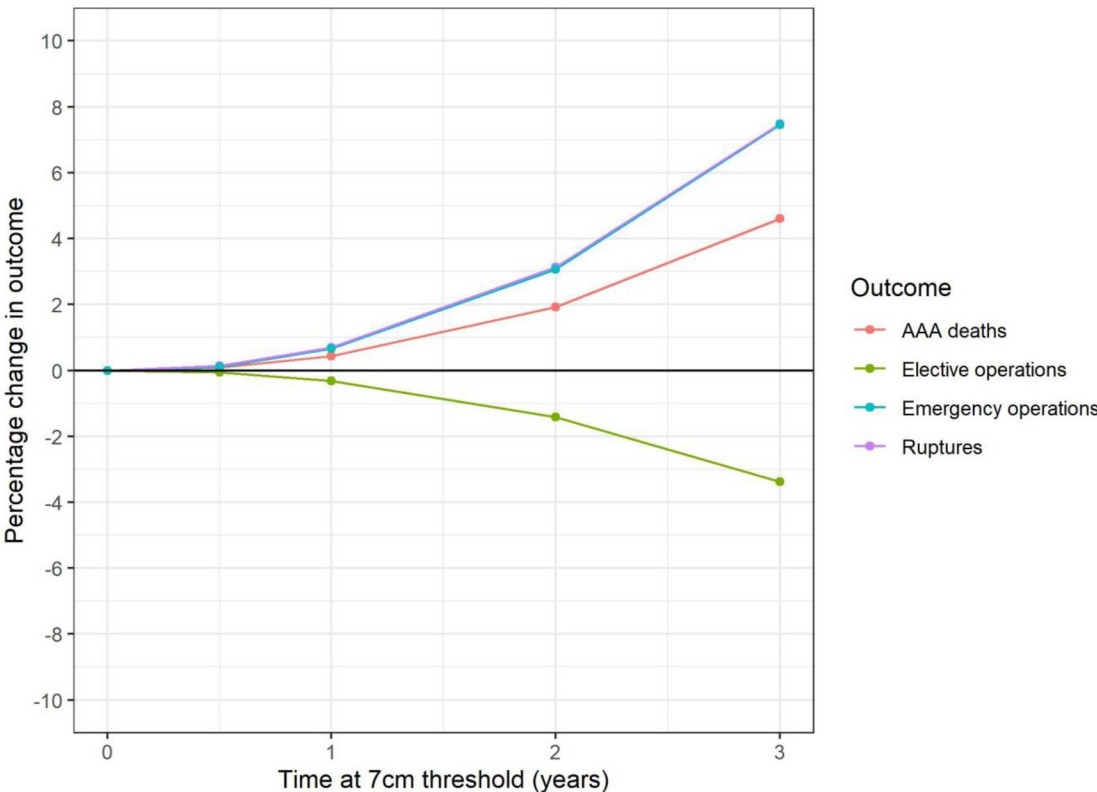

**Fig 5. Surveillance cohort: Change in key outcomes over varying time at increased (7cm) threshold (model I3).**

Table 2 the corresponding numbers of events for AAA deaths and emergency operations. The results suggest that the impact is additive, with a 10% increase in AAA-related deaths in a scenario with a one year suspension of surveillance scans combined with an increase to 10% drop-out/annum applied for two years and use of a 7 cm threshold for two years. These changes in combination have a similar impact on AAA-related deaths as a four year suspension of surveillance scans alone.

## Discussion

The results presented here suggest that short-term pauses in service provision following the COVID-19 outbreak in March 2020 are unlikely to have a large impact on AAA-related deaths. Specifically, scans in those with medium AAAs (4.5–5.4 cm) under surveillance should be resumed within one year, within two years in those with small AAAs (3.0–4.4 cm) under surveillance, and before two years in the invited group, and provided backlogs are cleared quickly. It might be expected that increasing the threshold size for surgery might have a more immediate and dramatic effect on AAA-related deaths. However, in March 2020, any men with AAAs ≥5.5cm had already been referred for elective surgery and are not included in this modelling of the surveillance cohort. Thus, there are no very large AAAs at the outset and it takes some time for AAAs in the 3.0–5.4 cm range to grow to a size where the risk of rupture (and AAA-related death) is very high.

The modelling further shows that any large drop-off in attendance could have a substantial impact on future AAA-related events. The results indicate that a drop to 65% attendance at the primary screening is approximately equivalent to a four year delay in the primary invitation

for the 65-year-old cohort in terms of increases in AAA-related deaths. Thus the more significant impact on clinical outcomes may not arise from service change policy, but from reluctance amongst these older men to attend scans during the transition period and being lost to follow-up thereafter. This suggests that there needs to be a concerted effort to ensure men who are due to be screened this year are strongly encouraged to attend screening following the COVID-19 outbreak and on resumption of services. As the national vaccination programme is rolled out, some 65-year old men may wish to postpone their primary screening invitation until after they have been vaccinated. Given the limited impact of delayed invitation in the short term, services can confidently defer the invitation, with re-invitation or encouraged self-referral of those men that have not attended during the lockdown and transition periods. Men on surveillance should be prioritised for vaccination, particularly those with medium AAAs who require an urgent re-scan [15].

Little other work has been carried out on the impact of COVID-19 on AAA services. A recent modelling study in the United States explored the trade-off between COVID-19 mortality and AAA-related mortality. The work focussed on the delay versus immediate repair decision for those with large AAAs, akin to considering an increased threshold for elective surgery [16]. The study uses a decision tree approach rather than the event simulation, rate-based approach used here. This necessarily simplifies the underlying processes; for example, the decision tree model does not include any modelling of AAA growth. The study concludes that the individual patient decision to delay or operate depends on surgical method, age, COVID-19 infection risk, length of delay (3-9m) and diameter. In contrast, we employ a long-term perspective and report results in terms of the impact on AAA-related mortality and surgical burden, scaled to the national population of England.

## Modelling assumptions

In addition to assumptions relating to the necessary simplification of the natural history of AAAs and to the estimation of transition rates and probabilities, this modelling work employs a number of additional assumptions related to its use in this context. Specifically, the model does not incorporate any reductions in capacity for screening/surveillance or intervention that may arise due to restrictions on staffing, social distancing or additional cleaning. It is assumed that delayed scans and interventions from the initial lockdown and transition periods can be accommodated immediately in later periods, as soon as services resume; longer modelled delays serve here as a proxy indicator for a slower resumption of services.

We have not modelled the risk of COVID-19 infection and related mortality. Estimates of nosocomial COVID-19 acquisition are not widely available, though published reports provide estimates ranging from 0.1% to 6% of admissions [17, 18]. However, these studies do not differentiate by length of stay, which is typically small for elective AAA repairs (2 and 7 days respectively for endovascular and open repair [19]). The National Vascular Registry in the UK reported 2.2% of those undergoing a vascular procedure tested positive for COVID-19 post-operatively, although this figure does not account for possible pre-operative community-acquired infection based on length of stay. Furthermore, these estimates of nosocomial transmission may not differ greatly from the risk of community-acquisition of COVID-19; in the period July-November 2020, the community incidence of COVID-19 in the UK ranged from 0.5 to 9.5 new cases per 10,000 per day (0.005% to 0.095% of the population per day) [20].

## Strengths and limitations

The discrete event simulation model used in this work is underpinned by detailed statistical modelling of AAA growth and rupture rates based on high quality data from a systematic

review and meta-analysis of growth and rupture rates of small AAA [21]. Furthermore, the DES has been well validated against data from the Multicentre Aneurysm Screening Study [2] and from a previous Markov model [22], producing reliable estimates of events over the trial follow-up. Data informing the diameter and age distribution of the national surveillance cohort were obtained directly from NAAASP and as such provide a realistic representation of this group for the modelling process.

There are a number of simplifications relating to model structure that were necessary for carrying out this COVID-19-related modelling work. The guidelines are intended for a short-term transition period, and the modelling results here assume that after this time, services will return to pre-COVID-19 levels for the remainder of the 30-year follow-up period tracked by the models. Additionally, the models also assume that any backlog in scans and elective interventions can be rapidly caught up. In practice, both of these aspects are likely to be more complex. The former can and has been explored in modelling, with delays to services well beyond the suggested period explored. However, the latter relates to capacity, which cannot readily be explored in this modelling setup. A backlog in surveillance scans, an increased waiting list for elective surgery together with reduced capacity arising from changes to cleaning protocols and/or staffing would necessarily result in further delays to scans and operations, and thus worsened clinical outcomes. Furthermore, the exclusion of men over 5.5 cm and already referred for surgery in March 2020 from the modelling means that any impact of service changes in this sub-group has not been included in the results presented here.

In addition to these structural assumptions, there are also challenges associated with extrapolating the underlying models of AAA growth and rupture rates to this setting. Specifically, these models are based on data acquired from studies of men with known AAAs under surveillance, where those with AAAs over 5.5cm in diameter generally undergo elective intervention unless they are considered unfit for this procedure. Thus estimates of growth and rupture amongst men with larger AAAs >5.5cm in diameter are based on extrapolations from model projections of men with small AAA (<5.5cm). In the modelling of post-COVID-19 policies here, more men reach these larger sizes, creating a greater influence of these more uncertain parameters in the model.

## Conclusions and implications for clinical practice

The relatively large impact of reduced attendance on clinical outcomes points to a careful consideration of the re-invitation policy relating to non-attenders in the surveillance programme during the pandemic. In the context of post-COVID-19 services, introducing future opportunities for scans for non-attenders at both primary and surveillance scans may help counter-act some of these effects. It may also be worthwhile to investigate strategies that may reassure and encourage attendance at both primary and follow-up appointments.

## Supporting information

**S1 Table. Input parameters used in the discrete event simulation model.**
(DOCX)

**S2 Table. Predicted excess AAA deaths and emergency operations in the national surveillance cohort over 30y period.**
(DOCX)

**S1 Fig. Convergence in the status quo model in the 65-year-old cohort.** Cumulative proportions for iteration numbers 100,000 to 10m: (a) AAA deaths, (b) ruptures, (c) elective

operations.
(TIF)

**S2 Fig. Convergence in the status quo model in the surveillance cohort.** Cumulative proportions for iteration numbers 100,000 to 10m: (a) AAA deaths, (b) ruptures, (c) elective operations.
(TIF)

**S3 Fig. Cumulative impact of scenarios on surveillance cohort.** S1 = surveillance scan suspension; S2.1 = 10% dropout/annum for 1 year; S2.2 = 10% dropout/annum for 2 years; S3 = 7cm threshold for 2 years before reverting to 5.5cm.
(TIF)

## Author Contributions

**Conceptualization:** Seamus C. Harrison.

**Formal analysis:** Lois G. Kim, Michael J. Sweeting.

**Software:** Michael J. Sweeting.

**Writing – original draft:** Lois G. Kim, Michael J. Sweeting, Seamus C. Harrison.

**Writing – review & editing:** Lois G. Kim, Michael J. Sweeting, Morag Armer, Jo Jacomelli, Akhtar Nasim, Seamus C. Harrison.

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
