## [Decision Letter · Decision Letter 0]

30 Apr 2021

PONE-D-21-09840

Modelling the impact of changes to Abdominal Aortic Aneurysm screening and treatment services in England during the COVID-19 pandemic

PLOS ONE

Dear Lois,

Thank you for submitting your manuscript to PLOS ONE. After careful consideration, we feel that it has merit but does not fully meet PLOS ONE’s publication criteria as it currently stands. Therefore, we invite you to submit a revised version of the manuscript that addresses the points raised during the review process.

We look forward to receiving your revised manuscript.

Kind regards,

Janet Powell

Academic Editor

PLOS ONE

Additional Editor Comments:

Thank you for this interesting submission. It has been reviewed by a modeller and two screening clinicians, one from the UK and one from Sweden. To make your paper more readily understood by screening and other clinicians, please address the comments of reviewers 2 and 3.

Journal Requirements:

Reviewers' comments:

Reviewer's Responses to Questions

**Comments to the Author**

1. Is the manuscript technically sound, and do the data support the conclusions?

Reviewer #1: Yes

Reviewer #2: Yes

Reviewer #3: Yes

2. Has the statistical analysis been performed appropriately and rigorously? 

Reviewer #1: Yes

Reviewer #2: Yes

Reviewer #3: Yes

3. Have the authors made all data underlying the findings in their manuscript fully available?

Reviewer #1: Yes

Reviewer #2: Yes

Reviewer #3: Yes

4. Is the manuscript presented in an intelligible fashion and written in standard English?

Reviewer #1: Yes

Reviewer #2: Yes

Reviewer #3: Yes

5. Review Comments to the Author

Reviewer #1: Congratulations on an excellent piece of work. I downloaded the AAA_DEX_model-master folder. I ran the R code. I examined the functions, models and inputs. I appreciated your coding, which is much better than my juvenile efforts. Everything worked as you described in your paper. I didn't have the time to go through all coding lines, so my caveat is that there might be errors.

I have comments that you might or might not find helpful. I think that you need not action any of these for your current manuscript, but they might change some of your thinking for future work. I will take the liberty of emailing you separately, after I have submitted this review, so you feel under no duress to implement any of the following.

a) Outcome The primary metric of any healthcare intervention is quality-adjusted life years. AAA repair and screening have no value if QALYs are unaffected. I assume you primarily report AAA-related mortality because of the apparent failure of RCTs of early vs later repair of AAA < 55 mm to increase survival (UK SAT etc), and similarly a limited effect of screening on AAA rupture, rather than overall survival i.e. perhaps you think that the readers of their paper would dismiss QALYs if they were the primary outcome. My email to you will propose that you use your expertise to address this gaping hole in the logic of scheduled AAA repair: until it is shown to increase QALYs all that scheduled AAA repair can achieve is the replacement of one mode of death (rupture) with another (commonly dementia, cancer, stroke, heart failure, pneumonia). This substitution cannot be assumed to be favourable. I would like to see all papers on AAA repair to consider survival and QALYs as the primary metric. Number of deaths from rupture does not suffice.

b) Other cause mortality The main determinant of QALY with and without scheduled AAA repair is survival with and without AAA repair, which I think is best represented by median life expectancy (although the optimist might consider time to 10% survivorship a better metric for them and the pessimist might consider time to 90% survivorship a better metric). The effect of AAA repair on median survival is most sensitive to variation in ‘death from other causes’, rather than variation in AAA mortality (mostly determined by diameter). You use annual mortality rates in sheet ‘Other cause mortality’ for men aged 65 years to 95 years: I am unsure what the source of these data were. The ONS triennial rates for men in England (or UK) do not quite correspond with the rates you used, but with minimal disparity (I think that you used the ‘qx’ rate rather than the ‘mx’ rate, with which I agree). At least, I did not spot a column in the last six ONS triennial releases that exactly corresponded with your rates. Mortality rates in all age groups and in both men and women have been decreasing over the past century, although rates have increased over the past 12 months, for instance 7917 excess deaths in the age group 65-74 years in the UK. This represents an absolute excess of 1 per 1000 per annum in this age range, whilst age ranges 75-84 and 84+ have increased by 6/1000 and 18/1000 per annum respectively. The conundrum with modelling future survival with and without AAA repair (including screening) is estimating ‘other cause’ mortality rates in the next 30 years. COVID will have killed more of the ‘sick and frail’ men aged 65 years: it is likely that there will be an abnormal reduction in mortality for the next 1-2 years, possibly longer, assuming the national effects of the pandemic resolve. After perhaps five years from the end of the pandemic one might assume that the average annual relative reduction in mortality of 1% to 1.5% might resume. I think that the effect of COVID on patients during hospital admission has much less effect on your model than the effect of COVID outside the hospital (although I haven’t modelled it). If one were going to be precise one could model the probability of COVID infection (possibly asymptomatic) before scheduled AAA repair and the suggestion that 7 weeks’ delay might reduce postoperative deaths to ‘normal’ (e.g. https://associationofanaesthetists-publications.onlinelibrary.wiley.com/doi/10.1111/anae.15464).

c) Heterogeneity of treatment effect My email to you concerns heterogeneity of treatment effect, in the main, which you did not model, but which is crucial for individual decisions and if done well would increase cohort benefit (and reduce harm). As I alluded to above, the effect of scheduled AAA repair on survival is most sensitive to rate of death from other causes: a man or woman with a median life expectancy of two years won’t benefit from AAA repair irrespective of risk of rupture (ie. any diameter); conversely, a man or woman with a much longer life expectancy has longer to accumulate risk of rupture, but more importantly the few deaths from rupture that occur whilst the aneurysm is ‘small’ result in a much greater loss of life and QALY than rupture of larger aneurysm in patients with otherwise short life expectancies. This heterogeneity in ‘other cause mortality’ makes any ‘risk of rupture’ threshold (mainly AAA diameter threshold) nonsense. I cannot avoid the logical conclusion that the decision to operate on intact AAA should depend upon the patient’s rate of ‘other causes of death’ as well as (or actually more than) the risk of AAA rupture. The offer of surgery should be triggered by a modelled increase in median life expectancy in excess of whatever threshold is worthwhile and affordable (maybe a year). I am not arguing against 55mm as a threshold, I am arguing against any threshold that uses AAA diameter alone. Incidentally, were a diameter threshold the correct metric for anyone, it wouldn’t be 55mm. I have modelled the UK SAT: the observed median life expectancy with earlier repair was 1.5 years longer than with later repair was replicated in simulation using the reported rates and times of repair in the two groups. The 55mm threshold is a consequence of UK SAT being underpowered for the particular outcome metric that they specified and the implementation of the protocol making management of the two groups making them more similar than one might suppose from the Methods.

As you know, the survival curves in the UK SAT are specific to that population in 1993. You can imagine what the result was when I simulated the UK SAT in 2020.

d) A general equation for aortic expansion If I interpret your input correctly you modelled an annual increase in aortic diameter of 6%. I appreciate that you are primarily concerned with aortic expansion for a few years after measurement at 40-55mm. Although aneurysmal aortas are reasonably considered ‘different’ to smaller aortas, I think that it is possible to generate a general equation for aortic expansion that would be consistent with measurement of abdominal aortic diameter from birth to the age of 100 years. I spent about one year developing a simulation for a male population born in 1945 and ‘ran it forwards’ to screening at the age of 65 years, using a single equation to determine aortic expansion with age, combined with five equations for rupture (to determine sensitivity). I used ONS general population survival (back to the 1980-2 triennium and then inferred back to 1945) to use with these equations, and I used the distribution of aortic diameter determined on US screening (the same as your sheet ‘AAA Size Distribution’). The following equation for annual expansion was consistent with the number of males surviving to the age of 65 and the distribution of aortic diameters insonated: ((0.000003*power(mm,3))+((0.0017*power(mm,2))-(0.05*mm)+0.45).

e) Equations for rupture Out of interest I compared with your equation for rupture one of the various equations that I’ve developed, and it gives similar results: (0.0000000000003*power(mm,6.2))+(0.0000003*power(mm,2.4))-(0.000029*mm)+0.00024

Reviewer #2: This is a very interesting modelling study examining the impact of COVID on the AAA screening services. It’s timely and relevant, with the only other study like this I know of being a much lower quality publication (reference 13).

However, the paper is complicated and at times can be difficult to follow. It needs simplifying and during revision the methods needs to explain: What data went into creating the model and does it reflect recent publications of NAASP data from post 2018; where do your parameters/presumptions come from and reference them clearly please; and I would seriously consider reducing the number of scenarios as we do have data on how things changed during the worst of the pandemic. The results need to clearly (and simplistically) present how you get to the cumulative impact for each model.

Introduction

1. Line 54 - 55. This should be referenced.

Methods

1. The relevant equator network quality checklist should be added. This may be STRESS for this type of study.

2. Line 81. This needs to be referenced.

3. Line 91. The references in 8 and 9 are from 2018. There are more recent publications on rupture rates from NAAASP. Are these reflected in the model? Please could this information be added. (I note a reviewer has brought this up before but your explanation from line 99 does not make it explicit that you have updated your model).

4. Table 1. Services have largely resumed. Why do you have so many changes from status quo model? The presumptions in these models are confusing and need explaining more clearly.

Results

1. Table 2. Please add column headings which explain the numbers, rather than the term ‘Model” as a heading.

2. Again, I’m confused here having read the preceding text in detail. What is the ‘Period change applied for’? Services have largely resumed and we know how long they were suspended or reduced for.

3. Line 184. Do we have information on drop out rate, or how much we expect it to go up? I’m not sure where these presumptions have come from.

4. Line 120. I’m not getting a good feel for the ruptures, deaths and operations required then cumulative impact as a result of your models. I wonder if this information for each model could be put in a summary table as it is really the crux of the whole study. The figures are of less use and could be appendixes.

Discussion

1. Line 299. You haven’t really looked at the implications for clinical practice. You could model resource requirement, cost etc but that may be beyond the remit of this study. I would limit your conclusions to the results you present in terms of a large excess mortality unless there is careful consideration on how to catch up and encourage men to attend.

Reviewer #3: Thank you for inviting me to review this manuscript. The aim of this paper is to combine knowledge on the NAAASP and the ongoing pandemic on the actual outcome for the invited 65 year old men.

General comments

There is a tsunami of COVID papers, but so far this is a new arena. It is very important for screening-oriented clinicians as well as healthcare providers in general, and the data are scarce.

Since the implementation of screening does carry a large impact on the rupture frequency in the society as well as control activities and number of elective vascular procedures, this data set really does support screening services and vaccination priority services with fresh thoughts!

Struck by the very strong case built through the model, now in April 2021, one could be tickled by the use of including the true story; some parts of the simulation model has actually now passed by and could be reported as “real life data”, did the IRL outcome mirror the model?

Overall really interesting paper, with strong methodology within the limitations of the model as always.

Specific comments:

Overall the authors use “we” too much. Please cut down.

Short title. Uncertain that “modelling changes to AAA screening” ; can be interpreted in different ways?

Abstract

Always prefer a clearly defined objective.

Methods. 3 “ we”…

Would suggest that the registrydata used are mentioned, and the modelling method.

More of aim than method here.

Used thresholds and rationals (5 cm?) is lacking.

Findings.

Why do you use 5 cm here. In the UK in general your used threshold of 5.5 in UL corresponds to almost 6 cm in CT. In most European countries the threshold is 5.5 on CT for treatment. So the choice 5 ? please motivate in text.

Please define Safety.

Introduction

The introduction leans a bit too much on references, which the reader should not have to look up in order to understand the paper.

P3 l 51. Not all repairs ? please define what was restricted for AAA; also was this not regional?

P3 line 63-66. Please present crude numbers since they are published.

Either in table or in text.

Startled by the use “stark”. Is this an English word ?

The aim is understood but could be formulated better (such as the methods text in the abstract) in order to stimulate readers with little experience to read the paper!

Method.

P 4 L 83.

Recalled ? Not the normal vocabulary for screening or surveillance.

Table 1. In the model you use: 7 cm.

Was this really the true threshold during the period; no 6 cm ? 6.5 ?

There is a vast difference probably in rupture risk.

Uncertain of which underlying rupture data you put into the model. Please present.

Results.

It would be fantastic, and interesting to see the Real world data; how was it then ; but understand if this not fits with the paper. It would be nice as a final on the result section.

The dataset is very interesting.

It is of course always interesting to wonder on a modeling of a “not successful screening man “ and a high achiever; meaning; a man that doesn’t come on the first screen invite. Comes on the second. Missed the checkups, turns-up after 5 years with a rupture; vs the “good guy” that comes at all invited occasions. It is highly presumable that the missing outs in the first cohort due to the combined effect of covid- non compliants then will be the dropouts afterwards; it not “new persons”. Does this effect the model ?

Discussion.

Very nice discussion to read and reflect on.

The text on backlog; should fit into the discussion; not only in limitations, since this really is the core critic on modelling; that you cant bring in all aspects of care.

6. PLOS authors have the option to publish the peer review history of their article (what does this mean?). If published, this will include your full peer review and any attached files.

Reviewer #1: **Yes: **John Bernard Carlisle

Reviewer #2: **Yes: **Chris Twine

Reviewer #3: **Yes: **Rebecka Hultgren

---

## [Author Response · Author response to Decision Letter 0]

27 May 2021

Reviewer #1: 

Congratulations on an excellent piece of work. I downloaded the AAA_DEX_model-master folder. I ran the R code. I examined the functions, models and inputs. I appreciated your coding, which is much better than my juvenile efforts. Everything worked as you described in your paper. I didn't have the time to go through all coding lines, so my caveat is that there might be errors.

I have comments that you might or might not find helpful. I think that you need not action any of these for your current manuscript, but they might change some of your thinking for future work. I will take the liberty of emailing you separately, after I have submitted this review, so you feel under no duress to implement any of the following.

We thank the reviewer for their supportive comments and for taking the time to explore the underlying modelling. We note the comment that this review has provided the comments below for future interest rather than requesting or recommending changes to the current manuscript. 

a) Outcome The primary metric of any healthcare intervention is quality-adjusted life years. AAA repair and screening have no value if QALYs are unaffected. I assume you primarily report AAA-related mortality because of the apparent failure of RCTs of early vs later repair of AAA < 55 mm to increase survival (UK SAT etc), and similarly a limited effect of screening on AAA rupture, rather than overall survival i.e. perhaps you think that the readers of their paper would dismiss QALYs if they were the primary outcome. My email to you will propose that you use your expertise to address this gaping hole in the logic of scheduled AAA repair: until it is shown to increase QALYs all that scheduled AAA repair can achieve is the replacement of one mode of death (rupture) with another (commonly dementia, cancer, stroke, heart failure, pneumonia). This substitution cannot be assumed to be favourable. I would like to see all papers on AAA repair to consider survival and QALYs as the primary metric. Number of deaths from rupture does not suffice.

The reviewer is right in taking the view that using AAA deaths and ruptures as outcomes here, this allows the paper to be more readily accessible and quickly understood. This paper focuses on the impact that the pandemic has had on AAA-related outcomes as this is the primary concern of the screening programme. QALYs are a useful output, particularly for cost-effectiveness analyses, though we are not carrying out such an analysis here (see responses to other reviewers below). Of course, when considering policy for AAA screening and intervention, we can only explore how to optimise these services – the question of how to then improve outcomes for other conditions subsequently incurred in a patient in whom AAA death is averted is another matter entirely. 

With regard to the 65-year-old cohort, QALYs are averaged over a very large population invited to screening, the majority of whom will not have an AAA; this means any average gains are necessarily very small. Furthermore, the majority of scenarios examined here consider only short-term changes to policy. This means that any additional events will occur at the start of the follow-up period, providing some intuitive sense of the impact on survival for the cohort in question.

b) Other cause mortality The main determinant of QALY with and without scheduled AAA repair is survival with and without AAA repair, which I think is best represented by median life expectancy (although the optimist might consider time to 10% survivorship a better metric for them and the pessimist might consider time to 90% survivorship a better metric). The effect of AAA repair on median survival is most sensitive to variation in ‘death from other causes’, rather than variation in AAA mortality (mostly determined by diameter). You use annual mortality rates in sheet ‘Other cause mortality’ for men aged 65 years to 95 years: I am unsure what the source of these data were. The ONS triennial rates for men in England (or UK) do not quite correspond with the rates you used, but with minimal disparity (I think that you used the ‘qx’ rate rather than the ‘mx’ rate, with which I agree). At least, I did not spot a column in the last six ONS triennial releases that exactly corresponded with your rates. Mortality rates in all age groups and in both men and women have been decreasing over the past century, although rates have increased over the past 12 months, for instance 7917 excess deaths in the age group 65-74 years in the UK. This represents an absolute excess of 1 per 1000 per annum in this age range, whilst age ranges 75-84 and 84+ have increased by 6/1000 and 18/1000 per annum respectively. The conundrum with modelling future survival with and without AAA repair (including screening) is estimating ‘other cause’ mortality rates in the next 30 years. COVID will have killed more of the ‘sick and frail’ men aged 65 years: it is likely that there will be an abnormal reduction in mortality for the next 1-2 years, possibly longer, assuming the national effects of the pandemic resolve. After perhaps five years from the end of the pandemic one might assume that the average annual relative reduction in mortality of 1% to 1.5% might resume. I think that the effect of COVID on patients during hospital admission has much less effect on your model than the effect of COVID outside the hospital (although I haven’t modelled it). If one were going to be precise one could model the probability of COVID infection (possibly asymptomatic) before scheduled AAA repair and the suggestion that 7 weeks’ delay might reduce postoperative deaths to ‘normal’ (e.g. https://associationofanaesthetists-publications.onlinelibrary.wiley.com/doi/10.1111/anae.15464).

Our main interest in this work is a comparison between the “status quo” (pre-covid AAA screening and intervention policy) and short-term changes to services implemented during covid, rather than the repair versus no repair analysis described here. Furthermore, since we hold other assumptions constant (including other cause mortality) across all scenarios, differences in key clinical outcomes should largely reflect differences in policy. In doing so, one assumption we make is that there is no additional risk of covid transmission during a hospital admission over and above the risk in the community. It does not seem unreasonable to expect this additional risk to be very low, and furthermore, there are very little data or high quality published studies that could inform such a parameter. 

c) Heterogeneity of treatment effect My email to you concerns heterogeneity of treatment effect, in the main, which you did not model, but which is crucial for individual decisions and if done well would increase cohort benefit (and reduce harm). As I alluded to above, the effect of scheduled AAA repair on survival is most sensitive to rate of death from other causes: a man or woman with a median life expectancy of two years won’t benefit from AAA repair irrespective of risk of rupture (ie. any diameter); conversely, a man or woman with a much longer life expectancy has longer to accumulate risk of rupture, but more importantly the few deaths from rupture that occur whilst the aneurysm is ‘small’ result in a much greater loss of life and QALY than rupture of larger aneurysm in patients with otherwise short life expectancies. This heterogeneity in ‘other cause mortality’ makes any ‘risk of rupture’ threshold (mainly AAA diameter threshold) nonsense. I cannot avoid the logical conclusion that the decision to operate on intact AAA should depend upon the patient’s rate of ‘other causes of death’ as well as (or actually more than) the risk of AAA rupture. The offer of surgery should be triggered by a modelled increase in median life expectancy in excess of whatever threshold is worthwhile and affordable (maybe a year). I am not arguing against 55mm as a threshold, I am arguing against any threshold that uses AAA diameter alone. Incidentally, were a diameter threshold the correct metric for anyone, it wouldn’t be 55mm. I have modelled the UK SAT: the observed median life expectancy with earlier repair was 1.5 years longer than with later repair was replicated in simulation using the reported rates and times of repair in the two groups. The 55mm threshold is a consequence of UK SAT being underpowered for the particular outcome metric that they specified and the implementation of the protocol making management of the two groups making them more similar than one might suppose from the Methods.

As you know, the survival curves in the UK SAT are specific to that population in 1993. You can imagine what the result was when I simulated the UK SAT in 2020.

We agree that this is an important area. However, this is a separate question regarding optimisation of AAA services (in the absence of COVID) by taking a more personalised approach to intervention decisions. The scope of this paper is rather different and investigates the impact of COVID-related changes to services as they are currently implemented. 

d) A general equation for aortic expansion If I interpret your input correctly you modelled an annual increase in aortic diameter of 6%. I appreciate that you are primarily concerned with aortic expansion for a few years after measurement at 40-55mm. Although aneurysmal aortas are reasonably considered ‘different’ to smaller aortas, I think that it is possible to generate a general equation for aortic expansion that would be consistent with measurement of abdominal aortic diameter from birth to the age of 100 years. I spent about one year developing a simulation for a male population born in 1945 and ‘ran it forwards’ to screening at the age of 65 years, using a single equation to determine aortic expansion with age, combined with five equations for rupture (to determine sensitivity). I used ONS general population survival (back to the 1980-2 triennium and then inferred back to 1945) to use with these equations, and I used the distribution of aortic diameter determined on US screening (the same as your sheet ‘AAA Size Distribution’). The following equation for annual expansion was consistent with the number of males surviving to the age of 65 and the distribution of aortic diameters insonated: ((0.000003*power(mm,3))+((0.0017*power(mm,2))-(0.05*mm)+0.45).

e) Equations for rupture Out of interest I compared with your equation for rupture one of the various equations that I’ve developed, and it gives similar results: (0.0000000000003*power(mm,6.2))+(0.0000003*power(mm,2.4))-(0.000029*mm)+0.00024

We are grateful that this reviewer has taken time to review the underlying model that we have referenced on github (line 111). The 6% growth rate is taken from one of these files, but refers to the mean percentage increase in diameter per annum; the growth model itself incorporates changes in growth rate according to diameter (full details available in the relevant file and accompanying references). 

Reviewer #2: 

This is a very interesting modelling study examining the impact of COVID on the AAA screening services. It’s timely and relevant, with the only other study like this I know of being a much lower quality publication (reference 13).

However, the paper is complicated and at times can be difficult to follow. It needs simplifying and during revision the methods needs to explain: What data went into creating the model and does it reflect recent publications of NAASP data from post 2018; where do your parameters/presumptions come from and reference them clearly please; and I would seriously consider reducing the number of scenarios as we do have data on how things changed during the worst of the pandemic. The results need to clearly (and simplistically) present how you get to the cumulative impact for each model.

The full detail of the model has been published elsewhere, which we reference in line 94 of the text. We have now clarified in the text that this reference refers to the model itself (line 92). All of the parameter inputs and sources are also given in the supplementary information accompanying this paper (Supp Table 1; line 97), with full references; details confirming dates have also been added. We have added a footnote to Table 1 in the main paper to highlight this. Age and AAA size distribution at baseline for the surveillance cohort were obtained directly from NAAASP in May 2020 (Suppl Table 1, line 118). AAA size distribution for the invited cohort comes from the first 700,000 men screened in NAAASP (2009-2014) giving an overall prevalence of 1.34%.

There are currently two scenarios relating to the invited cohort and four to the surveillance cohort. The majority of these report on the impact of short-term up to long-term (2-5y post-March 2020) changes in key parameters relating to service delivery. Even if the true covid-related parameter value (e.g. drop-out rate during March 2020 to March 2021) is known, the long-term impact of this change on clinical outcomes is not known, and that is what is investigated here. Furthermore, changes to underlying parameters such as drop-out rates may differ locally, so it is of interest to present results for a range of possibilities, as we do here.

Introduction

1. Line 54 - 55. This should be referenced.

References added to line 54 & 56.

Methods

1. The relevant equator network quality checklist should be added. This may be STRESS for this type of study.

The STRESS checklist for simulation studies has now been completed. 

2. Line 81. This needs to be referenced.

Reference added to line 82.

3. Line 91. The references in 8 and 9 are from 2018. There are more recent publications on rupture rates from NAAASP. Are these reflected in the model? Please could this information be added. (I note a reviewer has brought this up before but your explanation from line 99 does not make it explicit that you have updated your model).

There was a publication regarding rupture rates in AAAs <5.5cm in NAAASP published in 2019 (Oliver-Williams et al, Circulation 2019) and refers to follow-up for the years 2009-2017. However, this paper presents results for relatively wide categories of AAA diameter and furthermore the number of ruptures presented in the paper is low (n=31). Here, we employ a more sophisticated joint mixed modelling approach accounting for exact AAA diameter and growth. The model is fitted to data from 11 studies from the RESCAN consortium and uses multivariate meta-analysis to obtain pooled estimates of model parameters. Further information on this is provided in the amended footnote to Supplementary Table 1.

4. Table 1. Services have largely resumed. Why do you have so many changes from status quo model? The presumptions in these models are confusing and need explaining more clearly.

See response to earlier comment regarding reducing the number of scenarios. 

A footnote has been added to Table 1 to clarify that parameters not listed in the description of alternative scenarios remain unchanged from the status quo, together with a few notes of clarification in the body of the table.

Results

1. Table 2. Please add column headings which explain the numbers, rather than the term ‘Model” as a heading.

This has been amended as suggested for both Table 2 and Table 3.

2. Again, I’m confused here having read the preceding text in detail. What is the ‘Period change applied for’? Services have largely resumed and we know how long they were suspended or reduced for.

Column headings in Table 2 and 3 have been amended to improve clarity here.

3. Line 184. Do we have information on drop out rate, or how much we expect it to go up? I’m not sure where these presumptions have come from.

A relevant reference has now been added (lines 109, 194-6).

4. Line 120. I’m not getting a good feel for the ruptures, deaths and operations required then cumulative impact as a result of your models. I wonder if this information for each model could be put in a summary table as it is really the crux of the whole study. The figures are of less use and could be appendixes.

This comment refers to line 210, which refers to the start of the section on results of cumulative impact of changes. We agree that the numbers of events are also important: the results in Tables 2 and 3 provide the key figures (AAA deaths and emergency operations) corresponding to Figures 1-5. A similar table has now been added to the supplementary material relating to the cumulative impact results (Suppl Table 2).

Discussion

1. Line 299. You haven’t really looked at the implications for clinical practice. You could model resource requirement, cost etc but that may be beyond the remit of this study. I would limit your conclusions to the results you present in terms of a large excess mortality unless there is careful consideration on how to catch up and encourage men to attend.

There is already strong evidence supporting the cost-effectiveness of aneurysm screening pre-covid (see, for example, Glover et al, Br J Surg 2014). Since all of the scenarios modelled here represent a sub-optimal implementation of the NAAASP screening programme, the goals here are (i) to understand the potential long-term impact of changes that had to be made in light of the pandemic, and (ii) to highlight aspects of the programme which should be prioritised in the return to normal services in order to minimise adverse clinical events. We have therefore focussed our conclusions on these elements, with recommendations to reassure invitees at all stages of screening, and to ensure non-attenders are offered opportunities to return to surveillance (lines 310-315). 

An analysis of cost here would not provide the same insight as in a situation where policy decisions can be made based on cost and clinical effectiveness alone. A cost-effectiveness analysis here would likely show that reducing the surgical threshold back to 5.5cm is cost-effective, but cannot account for pandemic-related guidelines that may prevent this. We agree that modelling of limited resource use would be a useful addition, but – as the reviewer suggests – this is beyond the scope of this paper; we have discussed this as a limitation of this work in the manuscript (lines 291-297). 

Reviewer #3: 

Thank you for inviting me to review this manuscript. The aim of this paper is to combine knowledge on the NAAASP and the ongoing pandemic on the actual outcome for the invited 65 year old men.

General comments

There is a tsunami of COVID papers, but so far this is a new arena. It is very important for screening-oriented clinicians as well as healthcare providers in general, and the data are scarce.

Since the implementation of screening does carry a large impact on the rupture frequency in the society as well as control activities and number of elective vascular procedures, this data set really does support screening services and vaccination priority services with fresh thoughts!

Struck by the very strong case built through the model, now in April 2021, one could be tickled by the use of including the true story; some parts of the simulation model has actually now passed by and could be reported as “real life data”, did the IRL outcome mirror the model?

Overall really interesting paper, with strong methodology within the limitations of the model as always.

We are pleased the reviewer recognises the strengths and value of this work, and the importance of prompt publication. In response to the issue of some time-points in the modelling now being past, please see the reply to Reviewer 2’s first comment. 

Specific comments:

Overall the authors use “we” too much. Please cut down.

Some occurrences of “we” have been removed (lines 32, 117, 255, 267)

Short title. Uncertain that “modelling changes to AAA screening” ; can be interpreted in different ways?

Short title has been amended accordingly (line 22)

Abstract

Always prefer a clearly defined objective.

Methods. 3 “ we”…

Would suggest that the registry data used are mentioned, and the modelling method.

More of aim than method here.

Used thresholds and rationals (5 cm?) is lacking.

Use of NAAASP population data now added (line 35). Description of the modelling method (discrete event simulation) is in line 31. Unfortunately it is not possible to provide the rationale for all the scenarios within the constraints of the word limit for the abstract; the details of this are contained within the main text of the paper. 

Findings.

Why do you use 5 cm here. In the UK in general your used threshold of 5.5 in UL corresponds to almost 6 cm in CT. In most European countries the threshold is 5.5 on CT for treatment. So the choice 5 ? please motivate in text.

Since this section is describing results in those under surveillance at the start of the pandemic, they are necessarily under 5.5cm. This statement provides results in terms of excess deaths for the very largest of those under surveillance at that time, namely those 5.0-5.4cm. This has now been briefly clarified in the abstract (line 40); full details are in the main text (lines 173-176).

Please define Safety.

We now use the word “outcomes” here (line 41). 

Introduction

The introduction leans a bit too much on references, which the reader should not have to look up in order to understand the paper.

P3 l 51. Not all repairs ? please define what was restricted for AAA; also was this not regional?

Reference 1 provides an indication of change in overall admissions for aortic aneurysm (53% reduction in lockdown), though the reduction is likely to be more pronounced within the sub-group of elective rather than emergency repairs. It is also likely there are regional variations reflecting local descision-making, though these data (and our model) present results nationally. 

P3 line 63-66. Please present crude numbers since they are published.

Either in table or in text.

This information from reference 6 has now been added to the introduction (lines 64-66).

Startled by the use “stark”. Is this an English word ?

The aim is understood but could be formulated better (such as the methods text in the abstract) in order to stimulate readers with little experience to read the paper!

Please see above for changes made to the abstract. 

Method.

P 4 L 83.

Recalled ? Not the normal vocabulary for screening or surveillance.

We have used this terminology here to help distinguish invitations that are part of the surveillance programme from those that relate to the primary screening invitation. 

Table 1. In the model you use: 7 cm.

Was this really the true threshold during the period; no 6 cm ? 6.5 ?

There is a vast difference probably in rupture risk.

It is true that the rupture risk for 6cm v 6.5cm v 7cm AAAs differs significantly. However, because we are modelling only those men under surveillance in March 2020 (i.e. those <5.5cm), for scenarios with relatively short-term changes to the threshold, very few men will grow as large as 7cm in this period. In effect, this means that for short-term changes to the threshold, there will be little difference in the results from models applying 6cm, 6.5cm or 7cm thresholds. This is described in the discussion in lines 256-260. Here we use a 7cm threshold to reflect the NJVIB publication summarised in the Introduction, though there is also some evidence relating to the use of 7cm thresholds (see reference 6). 

Uncertain of which underlying rupture data you put into the model. Please present.

Details have been added to the footnote of Suppl Table 1.

Results.

It would be fantastic, and interesting to see the Real world data; how was it then ; but understand if this not fits with the paper. It would be nice as a final on the result section.

As discussed in the response to Reviewer 2, although it may be possible to acquire information about what did happen in practice in terms of these policies for the initial period following lockdown (and noting that relevant large-scale data are not yet published), we are interested here in the long-term impact of these policies on clinical outcomes, which of course is not available for comparison. 

The dataset is very interesting.

It is of course always interesting to wonder on a modeling of a “not successful screening man “ and a high achiever; meaning; a man that doesn’t come on the first screen invite. Comes on the second. Missed the checkups, turns-up after 5 years with a rupture; vs the “good guy” that comes at all invited occasions. It is highly presumable that the missing outs in the first cohort due to the combined effect of covid- non compliants then will be the dropouts afterwards; it not “new persons”. Does this effect the model ?

I think the question here is about double-counting in the surveillance cohort: those individuals who would have dropped out during follow-up are the same individuals who would now not attend due to COVID. Since we are still modelling drop-out that would usually occur in addition to non-attendance due to COVID, we may over-estimate non-attendance. It is an interesting point, though of course not readily explored or supported by data. The approach we take here is to explore a range of short-term increase to dropout rates, including no increase at all (which could be considered to model a scenario whereby those not attending due to COVID are the same individuals who would not have attended anyway). 

Discussion.

Very nice discussion to read and reflect on.

The text on backlog; should fit into the discussion; not only in limitations, since this really is the core critic on modelling; that you cant bring in all aspects of care.

We agree that the discussion around potential backlog of scans and operations is a critical one. However, since this is beyond the scope of the modelling carried out here, we have included this under the subheading of limitations, within the discussion section. 

---

## [Decision Letter · Decision Letter 1]

3 Jun 2021

Modelling the impact of changes to Abdominal Aortic Aneurysm screening and treatment services in England during the COVID-19 pandemic

PONE-D-21-09840R1

Dear Lois,

We’re pleased to inform you that your manuscript has been judged scientifically suitable for publication and will be formally accepted for publication once it meets all outstanding technical requirements.

Kind regards,

Janet Powell

Academic Editor

PLOS ONE

Additional Editor Comments (optional):

Thanks for responding carefully to the reviewer comments

Reviewers' comments:

Reviewer's Responses to Questions

**Comments to the Author**

1. If the authors have adequately addressed your comments raised in a previous round of review and you feel that this manuscript is now acceptable for publication, you may indicate that here to bypass the “Comments to the Author” section, enter your conflict of interest statement in the “Confidential to Editor” section, and submit your "Accept" recommendation.

Reviewer #2: All comments have been addressed

2. Is the manuscript technically sound, and do the data support the conclusions?

Reviewer #2: Yes

3. Has the statistical analysis been performed appropriately and rigorously? 

Reviewer #2: Yes

4. Have the authors made all data underlying the findings in their manuscript fully available?

Reviewer #2: Yes

5. Is the manuscript presented in an intelligible fashion and written in standard English?

Reviewer #2: Yes

6. Review Comments to the Author

Reviewer #2: Great changes.

Only comment is that putting the number of patients as well as % in the abstract woful help give context.

7. PLOS authors have the option to publish the peer review history of their article (what does this mean?). If published, this will include your full peer review and any attached files.

Reviewer #2: **Yes: **Chris Twine

---

## [Editor Report · Acceptance letter]

7 Jun 2021

PONE-D-21-09840R1 

Modelling the impact of changes to abdominal aortic aneurysm screening and treatment services in England during the COVID-19 pandemic 

Dear Dr. Kim:

I'm pleased to inform you that your manuscript has been deemed suitable for publication in PLOS ONE. Congratulations! Your manuscript is now with our production department. 

Kind regards, 

on behalf of

Dr. Janet Powell 

Academic Editor

PLOS ONE